# An Overview of Multiple Sclerosis In Vitro Models

**DOI:** 10.3390/ijms25147759

**Published:** 2024-07-16

**Authors:** Joanna Czpakowska, Mateusz Kałuża, Piotr Szpakowski, Andrzej Głąbiński

**Affiliations:** Department of Neurology and Stroke, Medical University of Lodz, Zeromskiego 113 Street, 90-549 Lodz, Poland; joanna.czpakowska@umed.lodz.pl (J.C.); mateusz.kaluza@umed.lodz.pl (M.K.)

**Keywords:** multiple sclerosis, in vitro, pluripotent stem cells, 3D culture, cell culture, cellular models, organoids, spheroids

## Abstract

Multiple sclerosis (MS) still poses a challenge in terms of complex etiology, not fully effective methods of treatment, and lack of healing agents. This neurodegenerative condition considerably affects the comfort of life by causing difficulties with movement and worsening cognition. Neuron, astrocyte, microglia, and oligodendrocyte activity is engaged in multiple pathogenic processes associated with MS. These cells are also utilized in creating in vitro cellular models for investigations focusing on MS. In this article, we present and discuss a summary of different in vitro models useful for MS research and describe their development. We discuss cellular models derived from animals or humans and present in the form of primary cell lines or immortalized cell lines. In addition, we characterize cell cultures developed from induced pluripotent stem cells (iPSCs). Culture conditions (2D and 3D cultures) are also discussed.

## 1. Introduction

Evidence of a neuroinflammatory condition called multiple sclerosis (MS) dates back to the 14th century, and this disease still poses a huge challenge in treatment [1]. The new approaches to researching the not fully understood pathology of MS and creating a therapy are desirable considering the fact that almost 3 million people are affected by this illness [1,2]. The majority are between 20 and 40 years old, which is the age at which people make choices considering important vital decisions, and MS influences them greatly [3]. The need for constant research can be especially supported by the statistic pointing out that 50% of MS patients are forced to function with a wheelchair [4].

The disease affects the gray and white matter in the central nervous system (CNS) [5] and mainly manifests in trouble with sight and hearing. Additionally, cognition is worsened. There are also difficulties with moving and accompanying fatigue [4]. These symptoms of MS are the consequences of spreading neurodegeneration that explains the presence of characteristic focal lesions in the CNS [5]. At the basis of this condition lies the increased permeability of the blood–brain barrier (BBB), which enables the immune cells to reach brain parenchyma [4]. This state initiates unfavorable consequences such as inflammation, loss of neuronal myelin with neurodegeneration, and the presence of gliosis [4].

In MS, both environmental and genetic factors, as well as lifestyle, contribute to the disease manifestation; thus, MS is named the ‘disease with a thousand faces’ with extreme heterogeneity in clinical course and pathomechanisms underlying disease development and course. Well-acceptable clinical division and characteristics of MS were proposed by Lublin et al. in 1996 and revised in 2013 [6,7,8]. According to this, MS is divided into the most frequent relapsing–remitting form (RRMS), which may turn into a secondary progressive form (SPMS), and the conversion usually occurs 10 years after disease onset. In addition, the primary progressive form (PPMS) is distinguished. The clinical classification depends on the presence of relapses in the disease course and the formation of lesions in white matter (active or non-active). More detailed characteristics have been described by Pitt et al. [8]. Heterogeneity MS phenotype manifests in its clinical course and depends on processes underlying MS pathology: acute inflammatory reaction, chronic inflammation, demyelination processes in white and gray matter, axonal degeneration, neuron loss, and remyelination [8]. These processes might be modified by various genetic and environmental factors and may be driven by various types of immune cells [9]. Heterogeneity in cellular activity and their molecular characteristic in the disease course generates the need to include this diversity and the role of particular cellular or molecular elements in MS research models.

## 2. The Cells of CNS Engaged in MS Pathology

The types of cells that play a crucial role in CNS during not only homeostasis but also unfavorable conditions such as inflammation are astrocytes, microglia, and oligodendrocytes (OLs) [1]. In the context of inflammation, microglia and astrocytes can modulate the extent of active inflammatory processes [1]. In addition, the immune cells impact the functionality of major CNS cells. Both sides can communicate due to the release of cytokines [10]. Microglia play the role of macrophages, which are characteristic of CNS and have immunomodulatory properties. Astrocytes are responsible for providing nutrients to the neurons and regulating the presence of neurotransmitters. The role of OLs in MS relies on being simultaneously the natural part of myelin and the source of autoantigenes [1]. In the context of inflammation taking place during the course of MS, OLs themselves cannot restore myelin, and those left are removed by astrocytes and microglia, escalating demyelination. In the case of a disrupted remyelination process, the next step is neurodegeneration. The damaged axon attracts macrophages in the active state, leading to secondary inflammation [10]. The consequence of the mentioned actions is present in the form of lesions; however, it is not clear how inflammation contributes to their emergence. The first step is thought to be either solely inflammation or neurodegeneration regulated by the immune reactions. Multiple studies indicate that the first option is the most accurate one, but it still is not certain [5].

## 3. Inflammatory and Demyelinating Processes in MS

The disrupted BBB favors inflammation in the CNS taking place according to the ‘outside-in hypothesis’, which is considered to be more accurate than the ‘inside-out hypothesis’ [2]. The ‘outside-in hypothesis’ comprises the activation of CD4+ T cells based on molecular mimicry in the peripheral circulation, which is followed by their entering the CNS, where antigen-presenting cells (APCs) such as B cells, dendritic cells, and macrophages come across them and initiate the mechanism of inflammation [1,2,10]. In addition, CD8+ T cells are numerous during the course of the disease, and they commonly affect myelin with CD4+ T cells, leading to its demyelination. At the basis of this process lies the recognition of autoantigenes such as myelin basic protein (MBP), myelin proteolipid protein (PLP), and myelin oligodendrocyte glycoprotein (MOG) [1]. The enduring inflammation, whose causes are still unknown, contributes to demyelination and, consequently, degradation of neurons [2]. The most visible consequences of these processes are lesions, which consist of fibrinogen and sustain chronic inflammation and demyelination [10].

In addition to the mentioned cause underlying MS, which is inflammation, demyelination also plays a crucial role in the onset of this disease. A few explanations for the presence of this phenomenon are suggested. One of them assumes that demyelination might be associated with the accumulation of autoantibodies directed against myelin antigens such as MOG and MBP [11]. Their action leading to demyelination comprises cytotoxicity toward myelin cells and their opsonization but also the onset of inflammation with the participation of macrophages and NK cells [12]. Another possible reason for demyelination occurrence is thought to be the presence of cytokines such as IFN-γ that lead to MHC expression in OLs [12,13] and TNF-α contributing to inflammation by acting on TNF receptor I [12,14]. The decreased number of connexins on OLs and astrocytes may also contribute to the process of demyelination that is visible in the worsening state of MS patients. Connexins are proteins that form a hemichannel in the cell and contribute to cell attachment by forming a gap junction that passes molecules smaller than 1.5 kDa and allows communication between cells. In the process of myelin shaping and maintenance, connexins between astrocytes and OLs are engaged. This is visible after the induced loss of gene expression for two major connexins proteins in OLs, which are Cx32 and Cx47. The lack of the first one leads to progressive demyelinating neuropathy in animal models, and loss of Cx47 is associated with myelin vacuolization. The importance of connexins such as Cx43 in MS can also be confirmed by the fact that their opening contributes to the induction of inflammation. This happens by increasing the amount of released chemokines and cytokines, which leads to astrocyte activation, and simultaneously, the Cx43 gap junction becomes a hemichannel. Elevated expression of the Cx43 protein is connected with the intensified presence of IL-17D and IL-33, which are involved in the attraction of the immune cells and lead to demyelination. Alternatively, the lack of Cx43 hemichannel activity results in remyelination, which is another confirmation that this protein contributes to the loss of myelin [15].

## 4. Neurodegeneration as the Consequence of the Inflammatory Process

The link between the activity of the immune system and its influence on neurodegeneration is especially visible in the production of substances such as reactive nitrogen species (RNS), reactive oxygen species (ROS), and nitric oxide (NO) by the active immune cells. They contribute to axonal dysfunction, which results in the inability to perform electrical conduction. Other targets of these products are mitochondria, whose damage results in the gradual loss of the source of energy and the death of neurons. The mitochondrial damage is especially threatening because the more organelle is damaged, the more oxygen radicals will be released, driving the destructive mechanism [16].

The other way of contributing to neurodegeneration relies on hypoxia, both genuine and virtual. The linkage between genuine hypoxia and loss of neurons can be explained by the fact that the presence of blood from arteries is limited at the specific sites of the brain, leading to the inhibition of oxygen and nutrient exchange. The cause of virtual hypoxia is abnormal mitochondrial functioning where the influx of Na^+^ takes place through their channels, and the simultaneous escape of unfavorable Ca^2+^ results in the activation of its canals [17].

Neurodegeneration can be intensified in patients of older age due to the accumulation of iron, which enhances the effect of oxidative injury. The initial neurodegenerative processes do not result in clinically visible dysfunctions, which are explained by the possibility of compensating lost neurons by the brain. However, over time, the progressive loss of neurons leads to decreased brain volume, which impairs its functions [17].

## 5. MS Etiology

### 5.1. Genetics

The confirmation that MS is partially determined genetically is derived from data indicating that the degree of kinship influences the risk of MS onset. Among the monozygotic twins, the risk rate is 25%, and for the more distant relative, the percentage drops. The part of chromosome 6 called the human leukocyte antigen (HLA) region consists of genes that are thought to be connected with many autoimmune diseases, including MS [18,19]. These are *HLA-DR2+*, *HLA-DQ6*, *DQA 0102* and *DQB1 0602*, *HLA-DRB1*, *DR15*, *DRB1*1501*, and *DRB1*1503*. The genome-wide association studies (GWAS) enabled the identification of genes with minor effects, which are *IL-2Rα* and *IL-7Rα*, making them the first not to be connected with HLA and simultaneously increasing the MS risk [18,19]. Generally, genes from this group improperly fulfill their main function, which is to control tolerance and regulatory mechanisms in MS [20]. The importance of HLA genes is especially visible while taking into account that carrying *DRB1*1501* allele is connected with a three times higher risk of MS. Furthermore, genomic analyses of data from the last 10 years confirm additionally the crucial importance of the immune system in MS development [19]. Despite the great importance of genetics in MS, it is suggested that this is a factor that contributes more to an increase in disease susceptibility than intensity. This is especially visible when comparing monozygotic twins, which have a significantly different course of the disease [20].

### 5.2. Environmental Factors

In addition to unfavorable genetics, the important role environmental factors such as vitamin D deficiency and tobacco smoking play in MS onset are thought to be the most essential. Other probable causes include obesity starting in childhood and infections [19]. The most important viral agents are the Epstein–Barr virus and human herpes virus type 6. On the other hand, considering bacterial agents, *Mycoplasma pneumoniae* can be distinguished [18].

#### 5.2.1. Vitamin D

In the context of emerging MS, vitamin D insufficiency is linked with the expression of genes that impact its regulation and simultaneously increase the risk of disease onset [21]. The other way of associating vitamin D with MS is through the immunological system and CNS because receptors for this vitamin are present on the surface of cells from both environments. T cells and B cells are additionally influenced by the lack of vitamin D, resulting in a block of their proliferation and consequently maturation, which in the case of T-helper 17 cells changes the profile of their formation and production of regulatory T cells starts being favored [22]. The importance of vitamin D in the prevention of MS onset is also visible in the process of remyelination based on the transition of oligodendrocyte progenitor cells (OPCs) to the OLs being part of myelin. Which is said to be boosted by vitamin D. Despite the many links between vitamin D deficiency and MS occurrence, the clinical trials assessing whether supplementation will improve the patient’s condition did not show consistent results, making it unclear if it is the cause of MS [23].

#### 5.2.2. Epstein–Barr Virus Infection

The association between the Epstein–Barr virus (EBV) infection and MS onset is visible in many studies. It has been indicated that the risk of MS development after such an event increases 32-fold in comparison to other infections, even if the route of viral transmission is similar [24]. This high score is even more surprising, taking into consideration that this common virus usually triggers mild infections. The strong connection between MS and EBV is detectable after the analysis of patients’ cerebrospinal fluid (CSF), where have been found oligoclonal bands reactive against EBV and antibodies for EBNA1 (Epstein–Barr nuclear antigen 1) and EBNA2 (Epstein–Barr nuclear antigen 2) epitopes. EBNA1 reactive antibodies were shown to cross-react with glial cell adhesion molecule (glialCAM) [25]. It is not understood yet in what way EBV contributes to MS onset, but it relies on the first contact with this virus and the immune system response. In the case of improper functioning, the accumulation of resident memory B cell and T cell follicles takes place in the specific parts of CNS where many pathologies begin [26]. The studies conducted on mice indicate the possible association of infection with EBV with MS onset by showing the cross-reactivity of EBV latent membrane protein 1 (LMP1) with monoclonal MBP-specific antibodies obtained from the CSF of MS patients. The induction of mice with LMP1 led to the production of autoantibodies reactive against myelin [27].

### 5.3. Contribution of Lifestyle to MS Onset

#### 5.3.1. Tobacco Smoking

Another substantial factor increasing MS susceptibility is tobacco smoking. Multiple case-control studies and surveys indicate the unfavorable effect of this stimulant. The relative risk for the disease onset is 1.5. Smoking can not only contribute to MS emergence but also exacerbate the disease course, making it more progressive [28]. This may happen due to the fact that they comprise many neurotoxins, such as cyanates, carbon monoxide, and free radicals. Additionally, regular tobacco smoking leads to chronic lung inflammation, which activates the immune system, and consequently, autoimmune diseases can emerge [29]. Currently, the accurate molecular processes leading to this state that are triggered by smoking are not known, and further studies are needed to elucidate them [28].

#### 5.3.2. Obesity

Regarding the risk factor, which is obesity, it was proven in many studies that its correlation with MS onset depends strongly on the age at which BMI indicates increased weight. Alfredson and Olsson indicated in their study that the periods of childhood and puberty are crucial for maintaining the proper weight because having a BMI > 27 at that time is linked with a doubled risk of developing MS in the future. Possibly, inflammatory processes accompanying obesity are responsible for the disease onset, taking into account that fat tissue secrete proinflammatory cytokines such as IL-6 and TNF-α [28]. The other inflammatory agents produced in higher amounts are adipokines favoring inflammation, such as leptin, resistin, and visfatin. Simultaneously, secretion of adipokines decreasing inflammation such as adiponectin and apelin is inhibited [30]. However, currently, the exact pathophysiological mechanisms have not been elucidated [28].

## 6. Importance of Developing the MS In Vitro Models

Each of the proposed factors that are thought to trigger MS onset still have unknown etiological mechanisms. Currently, it is not possible to indicate with certainty which of them should be considered more closely during research on MS causes. Additionally, the pathology of this disease comprises three major mechanisms (inflammation, neurodegeneration, and demyelination), and a specific explanation for the presence of each of them requires further studies. Nowadays, it is important to take steps towards establishing the specific targets for developing a novel MS therapy, taking into consideration epidemiological data clarifying how many people are affected by this disease and the fact that it happens mainly during the most important period of their lives. In vitro models of MS may turn out to be especially useful in this process by enabling researchers to elucidate pathological mechanisms and their causes. However, to fully benefit from the potential of in vitro models, it is crucial to develop the existing ones since their potential is restricted [31].

## 7. Comparison between In Vivo and In Vitro MS Models

To investigate the pathophysiology of MS, proper models of this disease must be utilized to mimic all of the processes responsible for its onset. The research on the patients’ tissues from the brain at the beginning of disease development is not possible due to its low accessibility. The alternative turned out to be the use of in vivo models, which led to a substantial increase in the level of knowledge regarding mechanisms in MS [31]. The animal models of the greatest importance that were developed reflecting probable causes of MS onset are experimental autoimmune encephalomyelitis (EAE), viral encephalomyelitis, and induced demyelination [32]. Unfortunately, this solution is not sufficient since the etiology of MS differs between humans and animals [31]. This is visible, especially in EAE, where it is not possible to avoid opportunistic infections coming from suppression or regulation of the immune system. Nevertheless, in vivo models are crucial in understanding the efficacy and safety of tested therapy because they provide an environment in which it is possible to observe the response of tissues and the whole organism [32,33].

In vitro models of MS might become the answer to the mentioned drawbacks. This approach to modeling MS provides strictly controlled conditions because of the applied simplification based on the assumption that only the specific types of cells are utilized. This approach prevents undesired influence from other cells, which takes place in in vivo models [32]. On the other hand, CNS cells function as part of the whole organism, which means that they must respond to the stimuli from their native environment. These signals influence their functioning, which will not be observed in the in vitro model, making the in vivo approach more desirable when the study is focused on investigating the whole MS landscape. The versatility of in vitro models also allows for the testing of tissues and mechanisms leading to injury, which is usually reserved for examination with animal models [31]. The use of in vitro models seems to be superior in comparison to in vivo models regarding the fact that there is no need to sacrifice many animals, which is considered an ethical approach (3R rule). The choice of utilizing in vitro models also turns out to be the cheaper option as the costs of maintaining animals are avoided, and the amount of reagents needed to conduct a study is decreased. Additionally, cell culturing provides a huge amount of replicable material that can be used in multiple repetitions, enabling reliable research outcomes [32].

## 8. Origin of Cells in MS In Vitro Models

Different sources of cells are taken into consideration when creating an in vitro model of MS. The first one is the mammalian brain, which provides primary cell cultures. CNS cells can be obtained from animals such as rats as well as from humans. Practically, acquiring cells from the human brain is not often performed because it is thought to be ethically questionable, which makes finding donors complicated. In addition, technically, the process of obtaining cells is difficult. In addition, further primary cell culturing is demanding and does not last long as their growth ends at a certain point. Similar difficulties are met regarding the rodent cells. Usually, they are obtained from rat embryos, where dominant OLs and neurons are found. Astrocytes and microglia come from the animal shortly after its birth. However, despite all of the weak points regarding primary cell cultures, they have a feature of paramount importance, which is their closeness to the conditions present in vivo [31,34].

Due to the fact that there is a huge discrepancy between human and rat nervous and immune systems and taking into consideration all of the mentioned drawbacks, a solution was proposed in the form of immortalized cell lines [31]. However, research on OLs and glial cells indicates that their functioning after immortalization may differ in comparison to the situation where they are situated in the organism. Additionally, the process of differentiation is not properly undertaken [34]. This was also visible in the study where immortalized neural stem cells (NSCs) were transplanted into a mouse brain. The cells did not display the capacity for developing fully mature neurons that are pre-synaptic or post-synaptic. The analysis of results also pointed strongly to the inhibited migration of these cells with simultaneously increased potential to proliferate, which may be advantageous considering the high number of cells needed for conducting research [35].

## 9. Types of MS In Vitro Models

Currently, several types of in vitro MS models are known. There are a few criteria that designate their division. The first one is focused on the origin of cells making up the model. There can be a distinction between primary rodent cells and primary human cells. Both divide into subgroups where the core of the in vitro model constitutes the specific type of CNS cells, which are usually neurons, microglia, astrocytes, and oligodendrocytes. In addition to the models based on primary cells, there also exist those made of human immortalized cell lines and their alternative, which are human CNS cell lines derived from induced pluripotent stem cells (iPSCs) (Figure 1) [34].

The need for the increased complication of structures mimicking the CNS environment in the in vitro models led to the development of more complex settings of cells. The natural organization of the CNS cells can be provided by brain organoids and brain slice cultures. Greater complexity can also be obtained by modified cell culturing. The most popular approaches are co-cultures, where at least two types of CNS cells and spheroids (3D cultures) are utilized (Figure 2) [31].

## 10. The Primary and Immortalized Cell Lines

### 10.1. Primary Neurons and Neuronal Cell Lines

Referring to the process of neurodegeneration that accompanies MS, it is crucial to take into account that neurons are one of the most important parts of the research into the use of in vitro models. The source of primary neurons in the case of animals is limited to rats and mice. The primary rodent embryonal hippocampus and cortex are widely utilized to obtain this type of cells [31,33]. Regarding humans, embryos and the post-mortem brain are considered to have the major origin. The first source, due to ethical considerations, is thought not to be an ideal solution [33,36]. In the case of brain tissues obtained post-mortem, the culturing of cells is not that efficient because proliferation does not take place in mature neurons. The cell cycle is not continued, which results in their death [33,37]. Obtaining neuronal cells during surgery is the other option, practiced less often [33]. Independently from the way of deriving the primary neurons, it is critical to isolate them from astrocytes and oligodendrocytes. Despite all of the mentioned drawbacks, the primary cell cultures are considered to be valuable because their origin provides that the majority of the neuronal features will be preserved [37]. Additionally, in the case of neurons obtained post-mortem, their great advantage is the knowledge regarding pathologies detected in donors and linked with genetic abnormalities, allowing an assessment of whether the outcomes of research conducted on those neurons are reliable [38].

In response to the weak sides of primary neuron culture, the shift towards neuronal cell lines has taken place [33]. In contrast to their alternative, which is primary neuron cultures, they distinguish themselves by their ability to grow continuously and uncomplicatedly. The cells from obtained cultures are characterized by a high level of similarity that makes them a perfect choice in the context of the replicability of the research results. The presence of all these features can be explained by the fact that the origin of the cell lines is neuronal tumors, which undergo immortalization in the next step. Unfortunately, they differ from the primary neuron in terms of the presence of aberrant dendrites and modified expression of mature neuron markers. To fully function as neuronal cells, there is a need for applying modification, such as utilizing an additional growth factor [31,37]. This is especially visible in the neuroblastoma cell line SH-SY-5Y, which needs retinoic acid. HCN and the NT2 cell lines have no markers of mature neurons, which means that they do not undergo differentiation. To obtain mature cells, stimulation with brain-derived growth factors is required [31,33].

### 10.2. Primary Oligodendrocytes and Oligodendrocytes Cell Lines

The main part of the myelin sheath constitutes OLs at various levels of maturity. Immature oligodendrocytes, pro-oligodendrocytes, mature oligodendrocytes, pre-progenitors, and progenitors can be distinguished. Only the mature OLs have the ability to produce MOG, MBP, PLP, and myelin-associated glycoprotein (MAG), which are essential proteins in the myelin sheath [31,33,39]. Their importance in providing myelin can be confirmed by the fact that one OL is able to make about 100 sheats for many neurons [40]. This feature distinguishes mature OLs from the other types. Oligodendrocyte cell lines and primary cells require specific conditions during culturing to become fully functioning Ols [31,33].

In view of the functions fulfilled by OLs, such as providing myelin and taking part in remyelination after an injury, it can be concluded that these cells would be useful in assessing their functionality in the context of MS, where demyelination is seen. Especially taking into account that in this disease, OLs, along with myelin sheath and its characteristic proteins, are first attacked by the immune system [31,33,41]. Since remyelination is a complex process involving the migration and differentiation of progenitors into the sites of injury, in vitro testing would be suitable to assess which modifications can enhance their activity. Additionally, modeling MS with the use of OLs allows for researching factors responsible for the damage of these cells and investigating the known ones, such as apoptosis and oxidative stress [31,33].

The medium used in oligodendrocytes’ primary cells and cell line culturing has a great impact on their further functioning, which is why choosing the proper conditions is crucial. Utilizing media without serum in culturing oligodendrocyte progenitor cells (OPCs) leads to their differentiation into OLs. However, the maturation in vitro appears to be not diverse, and the cytoarchitecture differs from that of in vivo cells [40]. The specific feature of primary OLs is that they are able to endure several weeks in vitro, which means that they last for a few passages during which they rapidly age [31,42]. Primary OLs can be obtained from rodents and humans. It is worth mentioning that the way OLs differentiate in animals is similar to the one in humans, which confirms that they share the same specific stages of this process [39]. In the case of primary human oligodendrocytes, they can be derived post-mortem and by biopsy. Alternatively, their source can be differentiated blood cells from the human umbilical cord and pluripotent stem cells. The process of OLs isolation relies on their lack of adherence to culture plates. The rest of the CNS cells remain attached to the surface, and OLs float on the surface after shaking, being accessible to isolation [31,33]. In addition, during this process, markers expressed characteristically for OPCs and mature OLs are also taken into consideration, by which these cells can be distinguished. In the case of OPCs, they are PDGFR-α and A2B5. OLs express O4 and GalC [41].

In response to the low yield of cells from primary cell culturing, limited access to animals, and ethical considerations, cell lines from animals and humans have been developed. Their main disadvantage lies in the process of their immortalization with the use of oncogene, which can influence cell functioning, and as a consequence, they do not reflect the behavior in vivo [41,42]. The most common examples are Oli-Neu from mice, OLN-93 from rats, and HOG derived from humans. Oli-Neu is created by transfection of primary cells with t-neu oncogene. The ability of this cell line to interact with neurons makes them an appropriate choice to assess the interaction between these two types of CNS cells. OLN-93 is created by the spontaneous immortalization of OPCs coming from mixed glial cultures from the brain. Culturing this cell line with a low amount of serum leads them to differentiate. The source of the HOG cell line is oligodendroglioma. Progenitors’ markers are present on their surface, and maturation markers can appear when serum levels are low [31,33].

### 10.3. Primary Microglia and Microglia Cell Lines

The activity of microglia in CNS is highly similar to functions of macrophages in the immune system [43]. They are also responsible for sustaining homeostasis and contribute to CNS development [44]. In the presence of detrimental factors such as inflammation, infections, and neurodegeneration, they become activated. However, the active microglia can be found not only in the area of demyelination but also in the properly functioning white matter [31,33]. The path leading to microglia activation determines their final state in which they become benign, protective, or detrimental. The involvement in neurodegeneration is conditioned based on their status [31]. In the case of acquiring detrimental properties, microglia may cause tissue injury and intensify neurodegeneration by releasing ROS, RNS, and pro-inflammatory cytokines [45]. In the context of MS, it is crucial to analyze their role in inflammation, as well as neurodegeneration, since they may be a proper target for modulating mechanisms taking place during these processes [33,46].

Primary microglia cells can be derived from animals such as mice, rats, and macaque or rhesus monkeys. It is also possible to obtain them from humans. Since the cells are coming from organisms in the embryonic or very early stage of life, they may not correspond with the nature of a chronic disease such as MS. Its characteristic features, such as neurodegeneration, appear in the later period of life; therefore, the way for eldering microglia cells would be desirable in the research on MS to fully reflect the prevailing conditions [33,45]. Ideally, microglia utilized for in vitro studies should be in the ‘M0’ state in which they are susceptible to encountered inhibitory signals such as transforming growth factor beta (TGF-β). This factor acts in an immunosuppressive way, which means that the cellular response to it increases the reliability of the in vitro model. Moreover, the transcriptomic studies on M0 microglia from various organisms indicated that the RNA transcriptome profile should be present in the optimal in vitro model [46]. Some studies on microglia from animals and humans showed that it is achievable with the use of cells derived during surgery or post-mortem. Additionally, the huge advantage of primary microglia cells acquired from affected brains is that they reflect the present pathways responsible for MS presence [32,33]. However, from an ethical point of view, the mentioned sources of microglia are not considered to be preferred [31]. The other drawbacks of the primary microglia are the low number of cells in the culture, which is not compensated by proliferation, and the rapid alteration of gene expression after isolation [44].

Primary cell lines can be superseded by microglia cell lines such as the HMO6 cell line obtained from the human fetal brain. HMO6 is generated after incorporation of *v-myc* oncogene with the use of a retroviral vector that leads to the expression of antigens characteristic for microglia (and also for macrophages) [31]. The other examples of microglia cell lines are MG6 and HAPI. The first one is derived from the brains of mice embryos and transformed with *c-myc*. The second cell line comes from the cerebral cortex of rats and undergoes a spontaneous immortalization procedure [46].

The process of microglia isolation is mainly based on their tendency to create a cell layer that can float and has no adhesive properties; therefore, the extraction is not complicated. In the case of microglia separation from a culture of primary glial cells, the features of astrocytes and oligodendrocytes are utilized. Astrocytes remain attached to the surface during the separation on a rotary shaker, and oligodendrocytes attach to them, while microglia form a separate fraction. It is important to mention that astrocytes must be approximately 100% confluent to avoid their presence among isolated microglia. The other ways of obtaining microglia are based on Percoll gradients, magnetic-activated cell sorting (MACS), fluorescence-activated cell sorting (FACS), and the limiting of nutrients [31,33,44].

The establishment of culture conditions for the isolated microglia is one of the most crucial aspects of maintaining this type of in vitro model since, after culturing in serum, they differ from microglia ex vivo and also from those coming from in vivo models which have been proven on the transcriptomic level [45,46]. Additionally, microglia in CNS have no contact with serum; therefore, they react differently, which is visible in their activated phenotype, which remains even after getting rid of serum. Responsible for this state are genes that are upregulated (*Apoe*, *Lyz2*, and *Lpl*) and downregulated (*Tmem119*, *P2ry12*, and *Cx3cr1*) in the culturing conditions. This fact restricts the spectrum of research that could be performed on microglia because other phenotypes are also important to investigate [47,48]. To optimize microglia models, there are approaches to resign from using the medium and utilizing the colony-stimulating factor1 (CSF-1) and also IL-34, which increase the chance for survival and intensify proliferation [43,46].

### 10.4. Primary Astrocytes and Astrocytes Cell Lines

The role of astrocytes in the unaffected CNS relies on providing favorable conditions to the rest of its cells. The fact that they account for approximately 50% of CNS cells indicates their importance in maintaining homeostasis, which is performed by regulating inflammatory processes and the amount of ions and neurotransmitters. Astrocytes are involved, especially in maintaining neurons and their synapses. Additionally, the astrocytic activity ensures that the BBB is formed correctly and remains tight [49]. Astrocytes also have the ability to influence BBB permeability and take part in allowing water and ions to pass through it [50]. In the case of emerging damage in CNS, they release growth factors such as platelet-derived growth factor (PDGF), insulin-like growth factor 1 (IGF-1), and basic fibroblast growth factor (bFGF), leading to intensified remyelination and simultaneously inhibition of OLs apoptosis [31,33]. However, since astrocytes can influence inflammation, they also can act unfavorably, causing degeneration by damaging axons and OLs. Additionally, astrocytes contribute to the intensification of damage if they undergo harmful processes that lead to BBB disruption since they are placed on 95% of the brain capillary on its side. After such an event, tight junctions stop fulfilling their roles, and leakage takes place. Characteristically, for chronic MS lesions, astrocytes form scars, and features such as hyperplasia with hypertrophy are present [31,33,50]. In the context of MS, their duality of reactions leads to the conclusion that the exact activity of astrocytes is still not fully elucidated. In vitro models made with the use of astrocytes enable us to analyze their responses to cytokines released during the MS course and additionally give the opportunity to test their reaction to some compounds, which could be potential future treatments. This is why research with astrocyte usage is desirable in discovering their involvement in MS [31,33,49].

The source of primary astrocyte cells encompasses the brains and spinal cords of animals such as mice and rats at the early stage of their lives, but in some cases, mature individuals are utilized. In addition, human brains are taken into consideration while obtaining primary astrocytes. This process can be conducted on embryos and adults post-mortem or during surgery [31,33]. The astrocytes derived from immature organisms are cultured in a medium with added fetal bovine serum (FBS), which allows them to proliferate and mature. However, the drawback of this approach is that maturation lasts weeks [49]. The alternative constitutes astrocyte cell lines obtained from astrocytomas in the case of humans. Considering mice and rats, cells are immortalized with the use of transfection. However, primary cell lines seem to be more desirable to make an in vitro model since their reactions to the tested factors would be more similar to those taking place in the native conditions [31,33]. It is important to mention that isolated primary astrocytes represent mainly the reactive phenotype, which is not fully useful in research where quiescent and naïve astrocytes reflecting the state of CNS in homeostasis are more desirable [49]. The crucial aspect of primary astrocyte isolation is that microglia are easily attached to astrocytes, and to make research outcomes reliable, and they must be separated. This can be achieved with the use of L-leucine methyl ester, which is toxic to microglia [31,33].

The refinement of the astrocyte in vitro model requires taking into consideration that these cells present in an organism are heterogeneous, and to recreate in vivo conditions in the model accurately, there are a few important issues that must be followed. First, if the research refers to either the brain or spinal cord, the astrocytes should be derived from the source that is of interest to the study. Second, if an assay requires the reactive phenotype, astrocytes should be at this state from the beginning, or stimulation should be performed. Last, to assess the cells’ structure with a focus on the cytoskeleton, channels to transport ions and neurotransmitters, flow cytometry and RNA sequencing should be performed [49].

## 11. Human CNS Cells Derived from iPSCs

### 11.1. MS In Vitro Models Based on iPSCs-Derived Human CNS Cells

The presented approaches to creating MS in vitro models are based mainly on primary cell cultures or cell lines. Unfortunately, there are limitations to these ways of mimicking the disease. In the case of primary cells obtained post-mortem, they allow for researching the pathology of the disease; however, they are not suitable for investigating molecular mechanisms that initiate MS [51]. The solution can be sought in iPSCs generated from somatic cells with the use of transcription factors such as OCT4, SOX2, KLF4, and c-MYC [51]. The huge benefit of using iPSCs is the fact that the cells can have a human origin, and then they are called human induced pluripotent stem cells (hiPSCs) [35]. The obtained cells are characterized by their two main features, potency and self-renewal, which means that they have the ability to differentiate into any other kind of cells from ectoderm, mesoderm, and endoderm. They simultaneously preserve patient-specific genetic composition that can be found in the progenitor cell. The ability is especially advantageous in the context of obtaining types of cells that are not easily accessible. The research on neurodegenerative diseases, neurons, and glial cells serves as an example [34,35,51,52,53]. The preserved genetic background can have significance in determining which variant of the particular genes can be involved in pathogenic mechanisms. Patient-specific iPSCs with the established phenotype can be utilized to create MS in vitro models [34,51]. The first attempt to create the iPSCs line from an MS patient took place in 2012. The donor was a female suffering from relapsing–remitting MS (RRMS). Since then, 52 such cell lines have been obtained, and the majority of them share similar features, such as origination from women aged 30–50 years with diagnosed RRMS [54]. This approach to investigating MS is especially beneficial, taking into consideration that a wide spectrum of genes associated with the immune system is involved in the pathology. Such models will allow for drug discovery and testing focused on the target in the environment mimicking disease, and they can also be useful in cell therapy [34,51].

The additional benefit of creating MS in vitro models with the use of iPSCs is that they allow adjusting conditions to evaluate certain aspects of the disease. In the case of neurodegeneration, utilizing iPSCs allows for assessing the response of neurons and also the rest of CNS cells to the toxicity factors. Based on the present genetic variant characteristic for a cell line in combination with the specific reaction for a toxin, it is possible to develop a personalized therapy. Addressing the immune response of CNS cells during the course of MS, it is also possible to research the interactions between them and T or B lymphocytes. iPSCs with an origin in one patient allow for differentiation of the cells from both the neurological and immune systems to reproduce the mechanisms present in vivo accurately [55]. It is possible to trigger neuroinflammation in cell cultures of microglia and astrocytes by adding a specific factor. In the case of microglia, LPS is utilized, which leads to an increase in the amount of IL-1β, IL-6, and IL-10 cytokines. The state of neuroinflammation can be present after the addition of IFN-γ or IL-1β. The addition of TNF to astrocytes results in the release of IL-6 and IL-8, which confirms the presence of inflammation [56].

### 11.2. Sources of iPSCs

The source of iPSCs can be blood, urine, and skin derived from MS patients. The isolated cells, such as fibroblasts, serve as a starting point in the process of differentiation (Figure 3) [34,57]. After the isolation of the desired cells, it is crucial to determine the method of cell reprogramming, with regard to the fact that this process is not efficient and the cells are destined for translational research, which means that the integration of sequences must be avoided. The most suitable methods turned out to be utilizing Sendai virus, episomal plasmids, and mRNA reprogramming [34]. In the case of obtaining iPSCs from fibroblasts of MS patients, a study conducted by Song et al. proved that this process delivers an adequate amount of cells with the proper morphology and gene expression for further usage, making this method relevant [57]. The alternative is urine-derived stem cells (USCs), which are considered to be superior to fibroblasts in terms of less complicated reprogramming, multipotency, and high rate of proliferation. From the practical point of view, obtaining USCs from urine is simple and non-invasive compared with a biopsy [58].

### 11.3. Differentiation of iPSCs into Neuronal Lineage

The first stage of iPSCs differentiation into neuronal lineage is called neural precursor cells (NPCs), whose markers are Mushashi 1, Nestin, GFAP, MAP2, and Muj1. The conversion can take place after iPSCs induction with TGF-β and by the addition of SMAD inhibitors, which ensure conditions similar to those during development [54]. This group of progenitors has the ability to become every one of the CNS cells after the application of the specific transforming factors described in the complex protocols. In the case of modeling MS in vitro, NPCs can be useful in researching the remyelination mechanism since they are found in sites of demyelination, enhancing the process of OL formation [34].

The differentiation of neurons from iPSCs is usually obtained according to protocols indicating that usage of factors such as N2, B27, BDNF, GDNF, and ascorbic acid, along with the proper culture medium, will mimic the conditions present during brain development [34]. In addition, in many of them, the standard procedure, such as dual SMAD inhibition, is performed [59]. This process ends successfully when obtained neurons have electrophysiological activity and form synapses [34]. There is also a possibility of directly obtaining neurons or NPCs from USCs, avoiding the need to create iPSCs. With the use of retroviruses or small molecules, the induction of differentiation leads to the emergence of fully functional neurons characterized by generating action potential and intensively outgrowing neurites [58]. In the context of MS, a study conducted by Song et al. resulted in obtaining neurons from MS patients’ iPSCs; however, they did not exhibit spontaneous action potential and synaptic activity, solely intensive resting membrane potential [57].

### 11.4. Differentiation of iPSCs into Astrocytes

In the case of astrocytes, their differentiation from iPSCs will allow us to research the neurodegenerative and inflammatory processes in the context of patient-specific genetic composition [34]. iPSC-derived astrocytes originate from NPCs, which undergo stimulation with Heregulin1b, IGF-1, and CTNF during astrogliogenesis. However, this process is time-consuming and complicated due to the necessity of sorting cells to obtain a homogenous population. The studies conducted in the field of differentiating astrocytes confirmed their functionality, showing that they can react to inflammation and release pro-inflammatory cytokines [34]. The possibility of obtaining astrocytes in the reactive state is crucial in the context of modeling MS in vitro because, in this state, they are favored by lesions present during the course of this disease [54].

### 11.5. Differentiation of iPSCs into Oligodendrocytes

The main components in researching processes such as demyelination and remyelination are OPCs and their mature, myelin-producing version—OLs. OPCs are obtained starting from patient-specific NPCs that are induced with PDGF-A, T3, NT3, IGF, and HGF. The further differentiation to OLs requires treatment with the glial differentiation medium [34,41]. Considering the novel methods for obtaining OLs, ectopic over-expression of transcription factors is worth mentioning due to its high effectiveness. This process relies on the transduction of NPCs with the use of a ‘SON construct’ based on the three characteristics for OLs transcription factors—SOX10, OLIG2, and NKX6-2 [42]. The possibility of obtaining the intermediate form, which is OPCs, allows us to investigate the obstacles that take place during the process of differentiation to OLs occurring during remyelination. Currently, the inability of OPCs to differentiate during the course of the disease is not fully elucidated; however, it is suggested that the lesion environment can have an influence [34].

### 11.6. Differentiation of iPSCs into Microglia

The last important element in creating the MS in vitro model is microglia in the reactive state [34]. Due to the fact that for an extended period, the origin of microglia remained elusive, this kind of CNS cell was the most complicated to differentiate [46]. Obtaining such cells from iPSCs is possible according to the five most common protocols. Generally, they are based on differentiating early mesodermal progenitors from iPSCs, and nextly, mature microglia are developed by activation of TGF-β and IL-34 pathways. Additionally, the common part of these protocols is the addition of CSF1 or IL-34, which enhances the chance of microglia survival [43]. The accuracy in differentiation of the obtained microglia was verified in multiple studies, indicating that on the transcriptome and functional levels, the similarity to native microglia is preserved. The acquired abilities, such as being responsive to inflammation and migration to the sites of demyelination, make them a proper choice for studying MS mechanisms [34].

### 11.7. Difficulties in Modelling MS with the Use of iPSCs

The presented promising way of modeling MS has several limitations, but they can be overcome by implying technical solutions. First, genetic variability is considered to be an advantage, but it can simultaneously pose a challenge in standardizing the outcomes from numerous cell cultures, each from a different donor. To prevent unexpected pathologies and reactions to the tested drugs, gene editing allows the setting of a uniform genetic landscape of cell lines leading to the creation of isogenic control lines [51,60]. Moreover, for the same reason, technical reproducibility is restricted [35]. The next drawback of utilizing iPSCs is the period of differentiation and maturation, which, in the case of neurons, vary among cell lines. The solution is the thorough recognition of each cell line’s characteristics and applying isogenic controls [51]. Additionally, CNS cells obtained from iPSCs, in fact, have never been functioning in their native environment, which in the context of modeling MS in vitro leaves room for discussion of whether their reactions to the tested factors will not differ from those in vivo. Considering the nature of MS, which is a neurodegenerative disease that emerges in older patients, it is important for the derived cells to preserve changes connected with aging, such as telomere shortening and being senescent. This can be achieved by the direct reprogramming of cells with the exclusion of obtaining the iPSCs phase [46].

### 11.8. iPSCs Cells in Modelling and Treatment of Other Neurological Diseases

Despite the complexity of neurodegenerative diseases, which are characterized by various genetic and environmental factors along with heterogenic etiology and more advanced age of patients, the modeling of neurodegenerative diseases with the use of iPSCs has proceeded successfully [35]. Contemporarily, it is possible in the case of Alzheimer’s disease (AD), Parkinson’s disease (PD), Huntington’s disease (HD), amyotrophic lateral sclerosis (ALS), and fragile X syndrome. The factor determining which disease will be modeled is either genotype or gene mutation. In the case of AD, different variants of *PSEN1*, *PSEN2*, *APP*, *APOE3/E4*, and *BMI1* genes are taken into consideration while preparing iPSCs for their usage in the in vitro model. Creating PD models relies on focusing on mutations in *SCNA*, *LRRK2*, *PINK1*, *PARK2*, and *GBA* genes. To create an HD model, characteristic CAG repeats in many different lengths must be taken into consideration while modeling this disease [51]. The essential part of ALS research is based on the differentiation of hiPSC into spinal motor neurons and also astrocytes with microglia to assess the inflammatory processes controlled by interactions between these cells [35].

Not only can the mentioned diseases be researched nowadays with the use of hiPSCs, but this technology also allows for the implementation of their treatments, which is confirmed by the clinical trials conducted. Generally, the main idea of treatment with the use of hiPSCs is based on the transplantation of human neural stem cells (hNSCs) into the sites with degenerated neurons [35].

## 12. Comparison of 2D and 3D Approach

### 12.1. 2D Culturing Method

Unquestionably, 2D cell culture has advantages. Adherent cell culture growing in monolayer conducted on culture plastic is easy to maintain and a low-cost method. Cell monolayer has a constant, no limited access to the culture medium, exchange of gases and waste products, and the same exposure to stimuli introduced in experiments. For this reason, the influence of the particular stimulus on cell functioning is easy to observe in contrast to tissue culture and culture in organoids, where the concentration gradient of stimuli and basal nutrients among tested cells is present [61]. The most important disadvantage of 2D cell culture is the lack of a third dimension, which causes several consequences. Microglia, as resident immune cells, and astrocytes, as cells involved in immune response and responsible for maintaining homeostasis, are especially sensitive to changes in the environment. Astrocytes and microglia in 2D culture show upregulated proinflammatory markers. The expression of *GFAP* and presence of vimentin in astrocytes, as well as chemokines and cytokines in microglia in vivo, is lower in comparison to these cells cultured in 2D [62,63]. Despite upregulated proinflammatory mediators, astrocytes and microglia are not activated and are capable of acquiring a full inflammatory phenotype [64,65]. There are many individual phenotypes between not activated (A0/M0) and activated (A1/M1) cells. Additionally, astrocytes in 2D culture show downregulated glutathione transporter 1, the important transporter involved in glutamate excitotoxicity resulting in neuron death [66,67]. There are prominent differences in the shape of cells. Astrocytes in vivo are similar to etoile, strongly branchy, and very complex, whereas, in 2D culture, they grow flat, round, or recall polygons. Microglia have an ameboid shape, whereas in vivo microglia have small-volume bodies with complex fine ramifications [68,69]. Both astrocytes and microglia acquire the ability to migrate almost exclusively during inflammation or tissue damage, i.e., when the glial scar is formed [70,71,72]. In traditional cell culture, both types of cells have low basal motility. Proliferation is an important subsequent feature that differentiates in vivo cells from 2D cultured cells. Astrocytes and microglia propagate relatively fast, which allows for convenient usage in experiments [72].

### 12.2. 3D Culturing Method

In the XXI century, there was a very large rise in interest regarding the subject of 3D cell culture, as seen in Figure 4. However, when comparing the number of publications with the phrase ‘3D culture’ to the number of publications with the phrase ‘cell culture’, 3D cultures still constitute a minor part of scientific research (data from Pubmed database). This simple example shows that although human tissues are spatially organized, current knowledge about cell biology is not based on 3D culture. Indeed, 2D cell culturing, most often conducted on flat glass or plastic dishes, has played a huge role in understanding cell biology and disease pathogenesis. Currently, the potential of 2D cell culture is slowly exploiting. However, cultures and co-cultures conducted in 3D enable studies about relationships between cells in more complex compositions and test drug candidates on models that more precisely mimic in vivo conditions [73]. The application of the 3D culture system led to the development of various approaches for in vitro modeling. The first one is cell biology-based models, which refer to spheroids and organoids. Their significant advantage is that the features characteristic for early moments of life formation can be found in such a setup. The second way of modeling is based on engineering. This approach allows for the creation of controlled conditions that will provide repetitive outcomes of research [74].

## 13. Cell Biology-Based Models

### 13.1. Spheroids

Spheroids are considered the most convenient way of forming the spatial environment for modeling MS in vitro. There is no need to utilize a scaffold to maintain the composition of cells—they are able to form themselves or with the aid of adhesion. This formation is possible after placing them in the hanging drop or with the use of non-adherent spinner flasks and centrifugation. The additional advantage of spheroids is that the extracellular matrix (ECM) composed of synthetic material does not have to be incorporated because it is already produced by spheroids, which is similar to in vivo conditions [74]. To assess the functionality of neurons cultured in the spheroids, patch-clamp recordings were analyzed by Dingle et al., leading to the conclusion that neuronal activity was present and these cells became part of the synaptic network [75]. The possibility of researching signal transduction in the neuronal network can help in elucidating the pathological mechanisms of diseases affecting CNS. The attempts to model neurodegenerative diseases were made towards AD and PD; however, the assessments that were conducted were mainly concerned with morphology and neuron survival. The crucial aspect, electrophysiological evaluation, remains to be developed [74]. In the case of MS, there was an attempt to investigate de- and remyelination with the use of the spheroid culture. In the study conducted by Vereyken EJ et al., it was possible to obtain aggregates of rodent brain cells that were responding to the stimulation with lysophosphatidylcholine (LPC), confirming the usefulness of this type of in vitro model [76]. The usage of pluripotent cells allows for the creation of stem-cell-derived human cortical spheroids (hCSs). This type of 3D arrangement is characterized by the presence of a wide range of neuroectoderm- and mesoderm-derived cell types and higher complexity, allowing for the obtaining of more reliable outcomes of research conducted with their use [77].

### 13.2. Organoids

Organoids are another type of 3D CNS cell organization, making up ‘mini-brains’. In comparison to spheroids, they distinguish themselves by the higher level of complexity that is provided by the usage of pluripotent stem cells (PSCs) or iPSCs and adult stem cells (ASCs), which self-organize into these structures. The brain organoids are characterized by the ability to differentiate and self-renewal, making them a solid and durable model similar to human tissues. The outer parts of brain organoids especially show great similarity in physiology and functionality to the tissues in vivo. Organoids not only shape into whole-brain formation but also have the ability to develop into a specific structure of the human brain, such as the cortex, midbrain, hippocampus, and others [74,78]. Additionally, their huge benefit is that they give the possibility to recreate conditions characteristic of the human brain in the process of development [79]. The culture conditions for organoids assume that cell aggregates formed into embryoid bodies (EBs) are floating in a medium containing Matrigel as the site of their embedding [74,79]. However, the components of this substance are not known entirely, and their usage leads to the organoids’ loss of stability. Despite the vast range of possibilities that give brain organoids, there is a need to develop a vascularisation system that will provide nutrient and oxygen exchange because necrosis on the inside of organoids is very common [74]. In the context of modeling neurodegenerative diseases, the obstacle in their usage is seen regarding their short period of existence in comparison to the time of disease duration. This means that this model would be appropriate for rare early-onset diseases. The lack of methods leading to organoids’ rapid aging makes them currently inappropriate to use for neurodegenerative disease research where age-related inflammation and pathological factors of aging are present [79,80]. However, a study conducted by Daviaud N et al. indicates that cerebral organoids obtained from iPSCs of MS patients provide a useful platform for studying this disease considering the preserved genetic background. The lack of blood vessels in brain organoids allows for the verification of the functionality of CNS cells that are separated from the immune system [81]. The high complexity of human-origin organoids also brings up ethical issues, such as uncertainty about whether they can have cognitive functions and the ability to feel [82].

### 13.3. Human Brain Slices Cultures

Among all of the possibilities of modeling MS in vitro, none of the mentioned examples mimic the complete landscape of the disease. Human brain slices are considered to be the most faithful option in presenting the physiological and morphological features thanks to the ability to keep the native shape of the brain along with the preserved system of blood vessels, synapses, and neuronal networks [83,84]. Moreover, the microenvironment of tissue remains, which gives more accurate results from the research. Human brain slices give the possibility to research the majority of cell types present in the brain at once and additionally allow for control of this system [84]. Neurons and glial cells present in these slices endure inside the organism for over a month and still react to the tested factors, which makes them a profitable option for researching neurodegenerative diseases [83]. Additionally, human brain slices are an appropriate platform for testing drugs and molecules altering neurotoxicity and neuroprotection [85]. This was presented in the study on Fingolimod, which led to the inhibition of demyelination in cerebellar slice cultures. This type of in vitro model can be utilized in research on MS [86]. The source of human brain slices is tissues undergoing surgery and donors post-mortem; therefore, they are not widely available and are not often utilized [83].

## 14. Engineering-Based Models

### 14.1. The Importance of the ECM

In humans and other multicellular organisms, most often, intercellular spaces are filled with the ECM [87]. The ECM in CNS is composed of polysaccharides such as glycoproteins (e.g., tenascin-R), glycosaminoglycans (e.g., heparin, hyaluronan), proteoglycans (e.g., neurocan and brevican) and proteins such as collagens. Interestingly, the ECM constitutes about 17–20% of the brain [88]. Components together create an elaborate and spatial net performing three basal functions. First, the ECM is a scaffold for cells and is responsible for the physical properties of particular tissue. Second, the ECM interacts with membrane receptors of cells, influencing cell morphology, proliferation, and gene expression. Interestingly, it has been shown that ECM interaction with cells’ integrins (group of membrane receptors) is crucial for cell survival. There is a high probability of cell apoptosis if the cell loses connection with the ECM [89]. Third, the ECM constitutes a physical and biological barrier for molecules and cells, which means that ECM has an influence on communication between cells and cell response during inflammation or under harmful conditions. Moreover, ECM composition changes are linked to the limitation of cells in terms of migration. Neuronal and glial cells are primarily responsible for the production of the ECM in CNS [88]. The ECM is characterized by the presence of large molecules (e.g., proteoglycans and hyaluronic acid) with the addition of other molecules, for example, laminins and tenascins [90]. Additionally, the brain is a uniquely soft organ. There are different parameters describing physical properties used to characterize tissues and materials mimicking the ECM, as well as plastics used for cell culture. One of them also used in the further part of this section is stiffness, which is given in one of the types of elastic moduli—shear modules. Shear modulus describes an object’s tendency to shear (the deformation of shape at constant volume) when acted upon by opposing forces and is a ratio given in Pascal (Pa). Measurements for humans have shown that brain stiffness is 0.1–1 kPa, muscle stiffness is 8–17 kPa, and plastic and bone stiffness is 2–4 GPa [91]. The high proportion of collagen 1 to other collagens guarantees high stiffness of tissue. In CNS proportion of collagen 1 to other collagens is low because the brain is soft tissue [92]. Despite a huge disproportion between brain stiffness and culture plastics, studies about neurodegeneration, inflammation in CNS, and autoimmunological diseases are performed primarily with used culture plastics or glass.

### 14.2. Biopolymers—Hydrogels

The goal of employing three dimensions in cell culture is to mimic the in vivo cell growth conditions provided via the ECM. Indeed, cells cultured in 3D conditions characterize a higher similarity to in vivo cells [93]. Systems in 3D work better in situations where the imperfections in 2D systems become too visible. These imperfections increase as the complexity of the model increases. Surely, complex experimental setups focus on cross-talking between cells, the immune response, functioning biological barriers such as BBB or Blood-Cerebrospinal Fluid Barrier, studies on the functioning of entire organs as well as diseases with elaborate etiology such as the one present in MS.

Biopolymers used to produce hydrogels may have natural or synthetic origins. Natural hydrogels are usually components of ECM, such as collagens or fibronectins. The natural origin of biopolymers entails low uniformity of the prepared hydrogels. To ensure the lowest variability between experiments and good data quality, synthetic polymers such as polyvinyl alcohol or polyethylene glycol are used. The mechanical properties of the hydrogels can be modified with great accuracy by using synthetic ingredients to produce hydrogels, which is a desirable feature in 3D cultures [72].

The choice of hydrogel should be preceded by the analysis of hydrogel’s ability to bind to cells. The addition of cell adhesion peptides (CAPs) or mixing with naturally functioning proteins is required if the structure of hydrogel disenables binding cells to the scaffold. Tripeptide sequence Arg-Gly-Asp (RGD) and pentapeptide Ile-Lys-Val-Ala (IKVAV) are the most frequently used CAPs [94].

Hydrogels are stored in liquid form; hence, before using a hydrogel, crosslinking is required. Crosslinking can be divided into chemical and physical. Chemical cross-linking involves the covalent bonding of individual polymer chains, resulting in the formation of a well-organized network. Physical cross-linking stabilizes the network through non-covalent interaction, e.g., ionic bonds. The cross-linking process may be toxic to cells; therefore, before choosing the hydrogel, the influence of cross-linking on cells should be considered. For instance, alginate requires physical cross-linking via the introduction of Ca^2+^ ions. When culturing astrocytes or neurons within alginate, both cells should be allowed to be sensitive to Ca^2+^ concentration change. The cross-linking can lead to abnormal cell signaling in neural cells and excitotoxicity in neurons [95,96,97].

### 14.3. Application of Hydrogels for Astrocyte and Microglia Culture in 3D

There is a certain tendency in the application of hydrogels in 3D astrocytes or microglia cultures. Three subjects seem to be the most explored by researchers. i.e., fabrication and optimization of the hydrogel construct, glial scar formation after brain injury, and 3D hydrogel culture as drug testing platforms [98,99,100,101]. The fabrication and optimization focus primarily on the reproduction of structures presented in the brain, such as neurovascular unit (NVU) and brain tissue (neurons–astrocytes–microglia), or providing conditions as similar as possible to in vivo for single cell types. As reported by Potjewyd et al., the most appropriate hydrogels for the NVU model are collagen type 1, gelatin, fibrin alginate, gellan gum hyaluronan, self-assembling peptide, elastin-like polypeptide, polyethylene glycol. For example, alginate is ionotropic; its elastic moduli hovers around 0.2–3.5 kPa, and it requires enrichment with CAPs or other modifications to cell binding. Their pros are the possibility of tunneling and adding CAPs and other factors to promote cell adhesion, albeit alginate requires printing into the support matrix and in the presence of calcium ions [97,99,102]. Furthermore, collagen, RADA-16 peptide, and alginate are chosen for the culture of astrocytes alone. Astrocytes in co-culture models are usually introduced into collagen hydrogel. In turn, the use of collagen, RDA-16 peptide, or graphene foam is reported for microglia [72]. The authors of these works describe the use of these biocomponents in 3D culture as the key to increasing knowledge in the basic biology of microglia or astrocytes, as well as broader functional units of the brain. Moreover, 3D models using hydrogels can significantly contribute to more accurate results at the first stages of drug discovery. For example, according to Beharry et al., cultures of human retinal endothelial cells (HRECs) and human retinal astrocytes (HRAs) on 3D hydrogel scaffolds better characterize the blood–retinal barrier and should be introduced as a model in pre-clinical studies [98]. Studies on the prevention or remodeling of glial scars are difficult to conduct in 2D. In 3D conditions, much better reproduction conditions are provided in vivo, both physically and in biology, e.g., astrocytes. Recently, Fang et al. proposed a model using astrocytes and a collagen hydrogel. They report that astrocytes cultured in this way can proliferate, grow, and form scar clusters. Additionally, the tissue model shrank by 4.5%, and the elastic modulus increased by almost four times, which may indicate a similar scarring process.

In the context of MS, 3D models are used primarily to imitate the BBB. For example, Kim W et al. used matrigel to co-culture astrocytes and brain endothelium. In this study, membrane functionality was demonstrated by treatment with rituximab (a chimeric monoclonal antibody against the protein CD20) used in the treatment of MS and anti-TfR antibodies. The proposed model blocked the passage of rituximab across the BBB, while anti-TfR antibodies crossed the BBB using receptor-mediated transcytosis [103,104]. Moreover, there are also attempts to mimic MS brain lessons. Until now, a 3D model based on hyaluronic acid has been described, focusing on mimicking the tissue stiffness of active and chronic MS brain lesions. The study showed that neural progenitor cells used to generate the 3D model have greater expression of myelin basic protein (MBP), hexaribonucleotide binding protein-3 (NeuN), and Ki67, and stiffer matrices promoted cell clustering. However, pro-inflammatory factors introduced into the model reduced the expression of MBP, NeuN, Ki67, and also cell clustering [105]. Therefore, 3D models using hydrogels can make an important contribution to the study of glial scar and brain diseases [106].

## 15. Improvements in Culturing: Blood–Brain Barrier Models, Co-Cultures, and Focus on Heterogeneity

The BBB is a structure in the form of a monolayer composed of endothelial cells that are part of blood vessels such as microvessels and capillaries. Its main functions rely on isolating the brain from harmful and toxic factors while simultaneously conducting nutrients and metabolite exchange [107,108,109]. The ability of BBB to pass only selected substances is possible thanks to its inter-endothelial junctions, which make it a tight barrier, and the presence of efflux pumps and metabolic enzymes. Taking into account the fact that the BBB has a huge impact on CNS functioning and is involved in MS pathology, it is reasonable in the process of creating an in vitro model of this disease to insert in cell cultures the construction mimicking this barrier [107,110]. There were attempts to make BBB models based on brain capillary endothelial cells, both from primary cells and immortalized [110]. It was established that to verify whether the obtained BBB model will fulfill its function, the tight junctions, presence of transport systems, and ability to shape a huge monolayer must be examined. This research is based on the assessment of the specific markers. Tight junctions can be studied based on the presence of Occludin, Claudin-5, and ZO-1. ABC transporters are characterized by P-gp, BCRP, and MRP occurence. SLC transporters are recognized based on GLUT-1, LAT-1, and MCT-1 expression. The most common BBB models from primary cells are Bovine Brain Microvascular Endothelial Cells (BBMECs), Porcine Brain Microvascular Endothelial Cells (PBMECs), Rat Brain Microvascular Endothelial Cells (RBMECs), and Human Brain Microvascular Endothelial Cells (HBMECs). Examples of BBB models from immortalized cell lines are BB19 from the human brain endothelium, RBE4 from rat brain microvessel endothelial cell line, b.End3 from brain endothelial cells of SV129 mice, and hCMEC/D3 cell line from human temporal lobe microvessel endothelial cells [107].

To fully unveil the BBB in vitro model’s functionality, co-culturing with other types of CNS cells, such as astrocytes, pericytes, microglia, and neurons, is required [107,108]. Most often, the Transwell culture systems are utilized in making the static BBB model; however, it is impossible with their use to mirror the 3D shape of the BBB and the impact of blood flow, which can be recompensated in dynamic BBB models [107,109]. Co-culturing not only involves BBB cells with other kinds of CNS cells but also configurations such as neurons—microglia and neurons—astrocytes are tested. The purpose of co-culturing microglia with neurons is based on microglia’s ability to regulate the immunological status of the environment of neurons, which contributes to their phenotype [111]. In the case of astrocytes—neurons co-culture, the first type of cells serves as a supportive entity for neurons, and the fact that only one astrocyte has contact with many synapses impacting their activity makes it reasonable to place them in one culture to acknowledge the brain mechanisms better [112].

CNS cells, especially astrocytes and microglia, present in MS lesions occur in various subpopulations. Therefore, it is worth taking into consideration their heterogeneity when setting up in vitro MS models. It is known that astrocytes at a resting state are heterogeneous and occur in nine various subpopulations, which vary in morphology and functionality. In addition, reactive astrocytes are heterogeneous in their characteristic. They are distinguished on the basis of transcriptomic and proteomic analyses such as single-cell genomics, spatial mRNA profiling, and highly multiplexed protein imaging. The results of such analyses will lead to determining the complex characteristics of the MS tissue and also CNS cells in homeostasis [113]. In the case of microglia, their heterogeneity is based on the division into protective and destructive states. It is crucial to deepen the knowledge about both forms since a therapy against MS targeted at the destructive microglia does not aim to decrease the population, but it should lead to a change in the phenotype into the protective microglia [114]. This knowledge can give an opportunity to create in vitro MS models in which the specific subpopulation will be utilized, making the results of the study more reliable [113,114].

## 16. In Vitro and In Vivo MS Models for Testing Treatment Strategies

It is difficult to reproduce the full image of MS and its types on a model that is not 3D and does not have the preserved structure of connections and spatial settings as in the in vivo conditions. Thus, 2D and 3D models are not able to mimic all MS features. The majority of in vitro models prove themselves great in imitating particular aspects of MS, such as the interaction between cells in a given condition, secretory activity, demyelination, and acute inflammation [42,115,116,117]. However, there are models that mimic BBB functions and are broadly used in drug screening and studies about BBB stability [97]. In addition, simple 2D and 3D cell cultures are appropriate platforms for drug screening thanks to high repeatability and relatively low breeding costs [72]. Considering the fact that organoids are characterized by much higher complexity than other 2D and 3D models, it is possible to associate organoid models with the specific types of MS. The examples are organoids created from hiPSC-derived neural precursor cells, and also accelerating oligodendrocyte differentiation with the use of SOX10-based protocol can be treated as a model of chronic active MS [118]. Moreover, the utilization of hiPSCs technology in the in vitro models allows for the development of personalized therapies and screening of potential therapeutics for their neuroregenerative and neuroprotective properties.

Animal and transgenic animal models, despite their shortcomings, are the best representations of MS models, including types of MS such as RRMS, PPMS, or SPMS. Animal models such as the cuprizone-induced demyelination model and EAE contributed the most to understanding the pathophysiology of MS. Complexity of animal models, the occurrence of all cell types involved in MS pathogenesis, and the ability to observe behavioral changes in animal models is currently undoubtedly the closest to what can be observed in humans; therefore, they seem to be the most appropriate platform for drug testing [60].

## 17. Conclusions

The huge variety of in vitro models allows for the most appropriate one to be chosen, focusing on the specific needs of research. In the context of MS, three main aspects, inflammation, demyelination, and neurodegeneration, can be studied extensively using in vitro models. Inflammatory processes can be researched by focusing on microglia and astrocytes, which are involved in the regulation of this state. With the use of OLs and neurons, it is possible to analyze the mechanisms of demyelination. The studies on neurodegeneration require the involvement of all of the types of CNS cells. This approach, assuming that the study is conducted on one specific type of cell, is advantageous considering the lack of influence from other cells, which takes place in in vivo models.

Another asset of in vitro MS models is the vast availability of their sources, types, and ways of culturing. CNS primary cells and cell lines can be derived from animal and human brains. There is also a possibility of differentiating iPSCs into any other type of CNS cell. The additional advantage of this approach is the fact that iPSCs can be obtained from various sources, such as blood, urine, and skin. The wide accessibility does not refer to primary cells, which are derived from the brains of donors post-mortem or during surgeries. The way of obtaining these cells makes them less desirable than iPSCs. Immortalized cell lines are easily accessible; however, the process of immortalization changes their morphology, which may affect the results of research conducted on their use. In this context, iPSCs seem to be the more appropriate option because, after their differentiation, the genetic composition of the donor is preserved. MS is thought to have a genetic predisposition, which means that cells derived from patients may differ in functioning and phenotype compared with those unaffected by the disease.

Modeling MS in vitro is proceeded mainly by utilizing a 2D approach, which is a convenient and uncomplicated solution. However, the outcomes obtained from the study and their use may be misleading since CNS cells cultured on a flat surface constitute the easily accessible target for the tested compounds. Such a situation does not take place in the native environment, which means that a more advantageous option would be creating conditions similar to those in vivo. In vitro models offer the spatial arrangement of cells in the 3D culture. Similar conditions are provided by in vivo models; however, acting on an in vitro model is more convenient. The types of 3D culturing encompass spheroids, brain organoids, and brain slice cultures. To make the outcomes of research with the use of in vitro models more reliable, modifications such as co-culturing and BBB implementation are also utilised. Moreover, 3D in vitro models offer a promising and complex tool in MS research; however, their advanced nature implies sacrificing more time and resources than 2D cultures.

Despite all of the advantages of in vitro MS models, they also have many limitations, such as the possibility of different cell responses in comparison to in vivo conditions. It is of vital importance to choose a model that is adequate to the assumptions of the study. In the case of analysis of the genetic etiology of MS, iPSCs will turn out to be the more appropriate option than primary cells of rodent descent. On the other hand, in the need to obtain repeatable results in a study where the response to some agent of one type of CNS cells will be checked, the immortalized cell lines would be the most suitable solution. Unquestionably, in vitro models are an appropriate tool to research new ways of perceiving MS, such as smoldering MS.

## Figures and Tables

**Figure 1 ijms-25-07759-f001:**
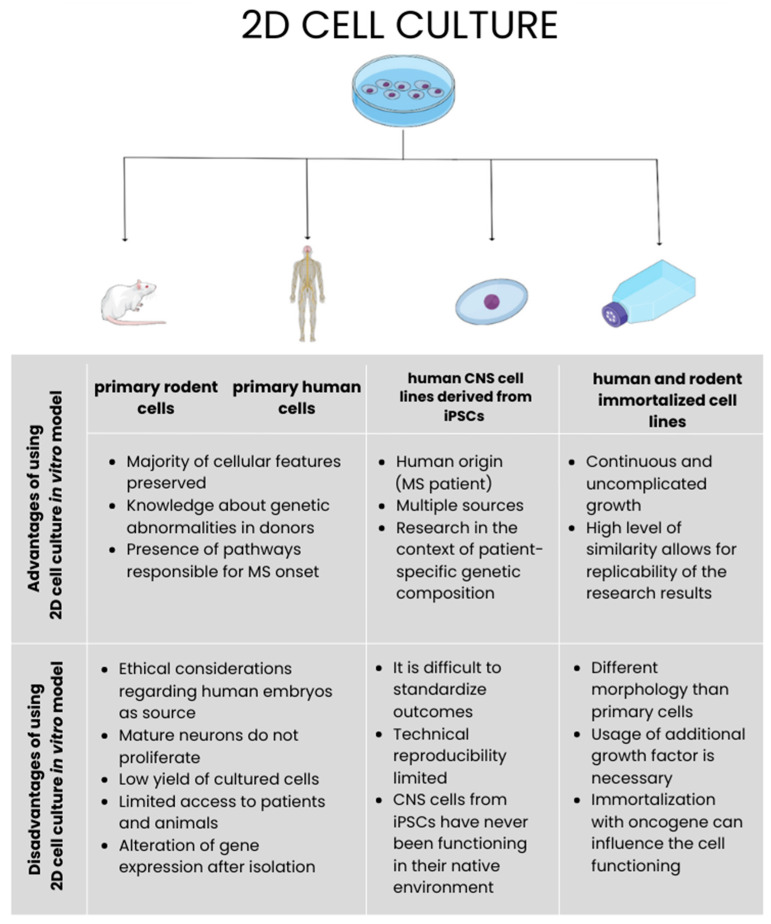
MS in vitro models based on 2D culture, with their advantages and disadvantages, allow us to decide which type of models will be the most appropriate for research on this disease.

**Figure 2 ijms-25-07759-f002:**
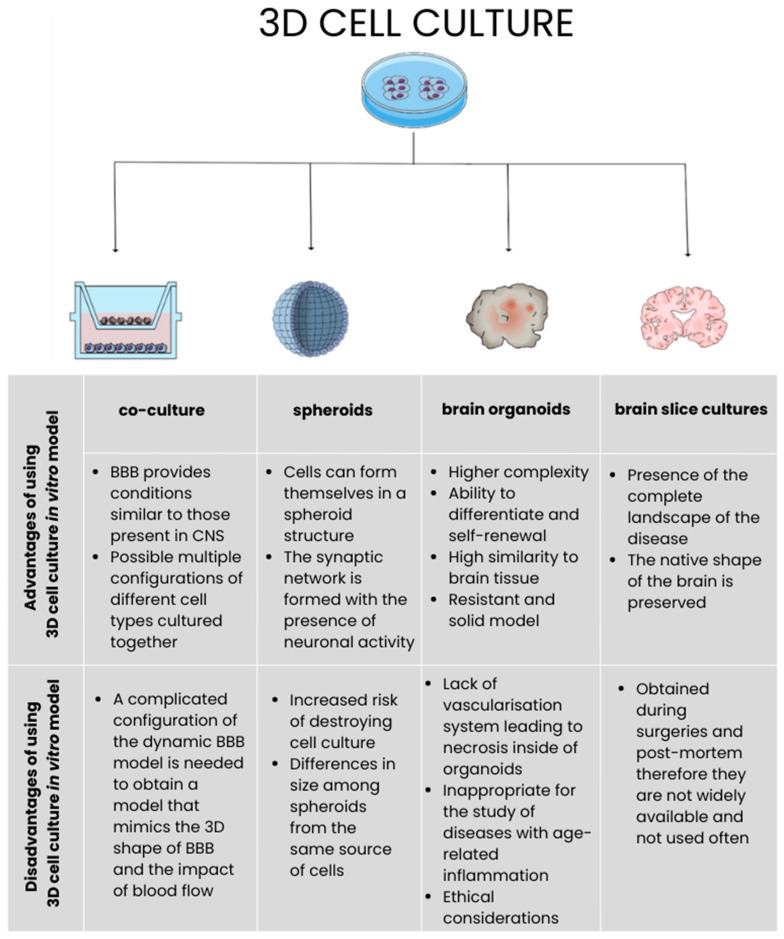
MS in vitro models based on 3D culture, with their advantages and disadvantages, allow us to decide which type of models will be the most appropriate for research on this disease.

**Figure 3 ijms-25-07759-f003:**
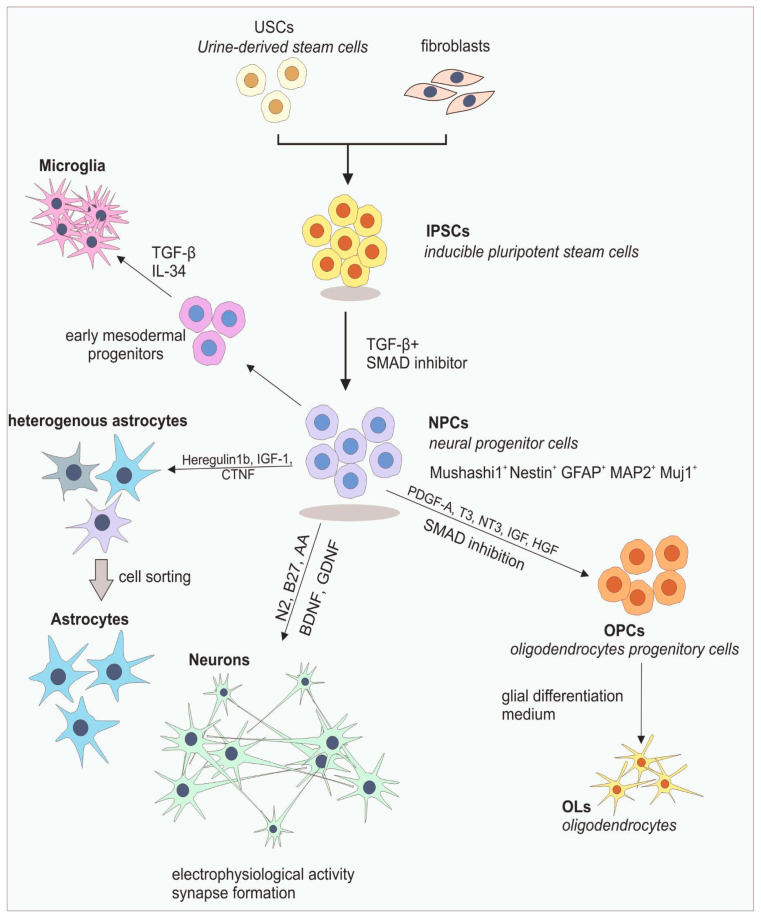
The sources of iPSCs, along with the methods of their differentiation into CNS cells.

**Figure 4 ijms-25-07759-f004:**
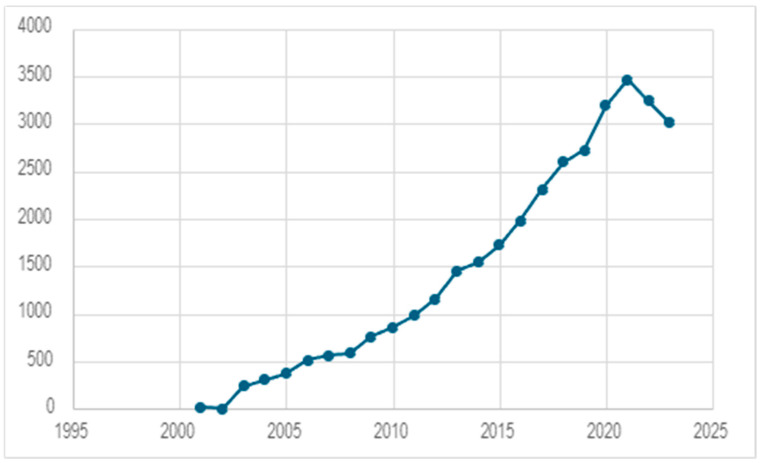
The number of publications with the phrase ‘3D cultures’ from 2001–2023 searched on the Pubmed database.

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
