# Peer review of "An Overview of Multiple Sclerosis In Vitro Models"

_ijms, 2024, doi:10.3390/ijms25147759_

Round 1

Reviewer 1 Report

Comments and Suggestions for Authors

Thank you for submitting this manuscript on in vitro models in MS.

This manuscript is interesting and well-written. It provides a lot of details on the current knowledge and discusses the different models.

I only have a few minor suggestions:

  1. I would suggest discussing why it is important to carefully choose the in vitro model and the potential consequences of choosing one that is not suitable.

  2. I would suggest discussing the complementarity of in vivo and in vitro models. When would an in vitro model be superior to an in vivo model, and vice versa?

  3. It would be really helpful if you could provide a decision tree about how to choose the right in vitro model.

Thank you 

Author Response

We are thankful for your review and precious suggestions. We hope that they will improve our paper. Please, follow our answers for more details.

Comments 1:

I would suggest discussing why it is important to carefully choose the in vitro model and the potential consequences of choosing one that is not suitable.

Response 1: 

Thank you for your suggestion, we have taken it into account it in the section ‘Conclusions’.

Comments 2:

I would suggest discussing the complementarity of in vivo and in vitro models. When would an in vitro model be superior to an in vivo model, and vice versa?

Response 2:

We have taken your suggestion into consideration and added the appropriate fragment in the section ‘Comparison between in vivo and in vitro MS models’.

Comments 3:

It would be really helpful if you could provide a decision tree about how to choose the right in vitro model.

Response 3:

This very helpful suggestion led us to decide to elaborate on Figures 1 and Figures 2 instead of providing a decision tree. We hope that the improved figures play a similar role to the suggested tree which is deciding about choosing the right in vitro model.

Reviewer 2 Report

Comments and Suggestions for Authors

Czpakowska et al. provides a nice paper describing the different in vitro models that can be used to study multiple sclerosis. The paper is useful for readers to pick the in vitro models that they would like to use in their studies and is broad audience. I recommend publication with the following comments addressed to improve the manuscript. 

(1)  Need more introduction on the different disease phenotypes and cell type heterogeneity.

Some key papers that might be useful:

https://www.ncbi.nlm.nih.gov/pmc/articles/PMC9324211/

https://www.neurology.org/doi/10.1212/NXI.0000000000200025

(2)  Some of the initial sub-sections may be shortened or combined to make the paper more concise.

(3)  Figure 1 and 2 may be improved to provide more useful details in terms of the usefulness of each type of models in MS studies.

(4)  There is a need to discuss how these in vitro models can model heterogeneity and provide insights to study heterogeneity using these models. For example, heterogeneity in microglia and astrocytes under different MS conditions.

For examples,

https://www.frontiersin.org/journals/cellular-neuroscience/articles/10.3389/fncel.2021.726479/full

(5)  What are some of the similarities and differences among these different in vitro models (primary vs iPSC differentiated cells or 2D vs 3D cells) – do they provide similar outcomes when using them in different MS studies? A discussion on this would be useful to the readers.

(6)  A table summarizing the different in vitro models would make it easy for the readers to pick the models that might be suitable for their studies.

(7)  More details on how each model with treatments can be used to study which type of MS progression (e.g., RRMS, PPMS, or SPMS) depending on the time course. If not, it would be good to provide a discussion on this, as opposed to the use of in vivo models.

Comments on the Quality of English Language

Some improvements in the English language may be required to facilitate on the flow of the reading.

Author Response

Thank you for your revision of our paper. We have followed your valuable suggestions. We hope they will make the paper even more useful for readers.

Comments 1: 

Need more introduction on the different disease phenotypes and cell type heterogeneity.

Some key papers that might be useful:

https://www.ncbi.nlm.nih.gov/pmc/articles/PMC9324211/

https://www.neurology.org/doi/10.1212/NXI.0000000000200025

Response 1: 

Thank you for your suggestion. We have added additional information in the ‘Introduction’ section.

Comments 2: 

Some of the initial sub-sections may be shortened or combined to make the paper more concise.

Response 2:

We agree with your suggestion therefore we modified the section ‘MS etiology’ where we combined some subsections to make the paragraph more concise.

Comments 3:

Figure 1 and 2 may be improved to provide more useful details in terms of the usefulness of each type of models in MS studies.

Response 3:

We followed your suggestion therefore we modified the figures by adding the additional information in tables where we show advantages and disadvantages of particular in vitro models which will be helpful in choosing the most appropriate and useful one in the conducted research.

Comments 4:

There is a need to discuss how these in vitro models can model heterogeneity and provide insights to study heterogeneity using these models. For example, heterogeneity in microglia and astrocytes under different MS conditions.

For examples,

https://www.frontiersin.org/journals/cellular-neuroscience/articles/10.3389/fncel.2021.726479/full

Response 4:

We have added additional information in section ‘Improvements in culturing: blood-brain barrier models, co-cultures, and focus on heterogeneity’ according to your suggestion. We have also taken into account the papers recommended by the Reviewer.

Comments 5:

What are some of the similarities and differences among these different in vitro models (primary vs iPSC differentiated cells or 2D vs 3D cells) – do they provide similar outcomes when using them in different MS studies? A discussion on this would be useful to the readers.

Response 5:

We have elaborated on this topic in the section ‘Conclusions’.

Comments 6:

A table summarizing the different in vitro models would make it easy for the readers to pick the models that might be suitable for their studies.

Response 6:

We introduced the mentioned tables as the improvements of Figure 1 and Figure 2. We hope that presented information will be more useful and clear for readers.

Comments 7:

More details on how each model with treatments can be used to study which type of MS progression (e.g., RRMS, PPMS, or SPMS) depending on the time course. If not, it would be good to provide a discussion on this, as opposed to the use of in vivo models.

Response 7:

Thank you for this valuable suggestion, we introduced a new paragraph ‘In vitro and in vivo MS models for testing treatments strategies’ in which we are elaborating on the proposed topic.